# PIK3C3 Inhibition Promotes Sensitivity to Colon Cancer Therapy by Inhibiting Cancer Stem Cells

**DOI:** 10.3390/cancers13092168

**Published:** 2021-04-30

**Authors:** Balawant Kumar, Rizwan Ahmad, Swagat Sharma, Saiprasad Gowrikumar, Mark Primeaux, Sandeep Rana, Amarnath Natarajan, David Oupicky, Corey R. Hopkins, Punita Dhawan, Amar B. Singh

**Affiliations:** 1Department of Biochemistry and Molecular Biology, University of Nebraska Medical Center, 985870 Nebraska Medical Center, Omaha, NE 68198-6125, USA; balawant.kumar@unmc.edu (B.K.); rizwan.ahmad@unmc.edu (R.A.); sai.gowrikumar@unmc.edu (S.G.); mark.primeaux@unmc.edu (M.P.); punita.dhawan@unmc.edu (P.D.); 2Department of Pharmaceutical Sciences, College of Pharmacy, University of Nebraska Medical Center, Omaha, NE 68198-6125, USA; swagat.sharma@unmc.edu (S.S.); david.oupicky@unmc.edu (D.O.); corey.hopkins@unmc.edu (C.R.H.); 3Eppley Institute for Cancer Research Program, University of Nebraska Medical Center, Omaha, NE 68198-6125, USA; sandeep.rana@nih.gov (S.R.); anatarajan@unmc.edu (A.N.); 4Fred and Pamela Buffett Cancer Center, University of Nebraska Medical Center, Omaha, NE 68198-5870, USA; 5VA Nebraska-Western Iowa Health Care System, Omaha, NE 68105-1850, USA

**Keywords:** autophagy, 5-FloroUracil, PI3KC3, chemoresistance, cancer stem cells

## Abstract

**Simple Summary:**

Colorectal cancer (CRC) represents a heterogeneous population of tumor cells and cancer stem cells (CSCs) where CSCs are postulated to resist the chemotherapy, and support cancer malignancy. Eliminating CSC can therefore improve CRC therapy and patient survival; however, such strategies have not yielded the desired outcome. Inhibiting autophagy has shown promise in suppressing therapy resistance; however, current autophagy inhibitors have failed in the clinical trials. In the current study, we provided data supporting the efficacy of 36-077, a potent inhibitor of PIK3C3/VPS34, in inhibiting autophagy to kill the CSC to promote the efficacy of colon cancer therapy.

**Abstract:**

**Background:** Despite recent advances in therapies, resistance to chemotherapy remains a critical problem in the clinical management of colorectal cancer (CRC). Cancer stem cells (CSCs) play a central role in therapy resistance. Thus, elimination of CSCs is crucial for effective CRC therapy; however, such strategies are limited. Autophagy promotes resistance to cancer therapy; however, whether autophagy protects CSCs to promote resistance to CRC-therapy is not well understood. Moreover, specific and potent autophagy inhibitors are warranted as clinical trials with hydroxychloroquine have not been successful. **Methods:** Colon cancer cells and tumoroids were used. Fluorescent reporter-based analysis of autophagy flux, spheroid and side population (SP) culture, and qPCR were done. We synthesized 36-077, a potent inhibitor of PIK3C3/VPS34 kinase, to inhibit autophagy. Combination treatments were done using 5-fluorouracil (5-FU) and 36-077. **Results:** The 5-FU treatment induced autophagy only in a subset of the treated colon cancer. These autophagy-enriched cells also showed increased expression of CSC markers. Co-treatment with 36-077 significantly improved efficacy of the 5-FU treatment. Mechanistic studies revealed that combination therapy inhibited GSK-3β/Wnt/β-catenin signaling to inhibit CSC population. **Conclusion:** Autophagy promotes resistance to CRC-therapy by specifically promoting GSK-3β/Wnt/β-catenin signaling to promote CSC survival, and 36-077, a PIK3C3/VPS34 inhibitor, helps promote efficacy of CRC therapy.

## 1. Introduction

Despite significant advances in its clinical management, colorectal cancer (CRC) remains the second leading cause of cancer-related death [1]. As per recent estimates, approximately a million new colon cancer cases are reported worldwide every year, and about half of these patients die from the disease [2]. The five-year survival for CRC patients depends largely on the disease stage as it ranges from 90% for patients with localized disease (stage I) to ~10% when the tumor becomes distant and invasive, and spreads to distant organs (stage IV) [3,4]. The basis of the advanced CRC treatment consists primarily of chemotherapy; however, drug resistance remains a major hurdle [5]. Thus, there is an urgent need for novel therapeutics that can help overcome chemotherapy resistance and restrict CRC progression.

Cancer stem cells (CSCs) are a specific cellular subpopulation within a cancer and possess the capacity to self-renew and generate cancer cell lineages [6]. The CSCs play a central role in cancer progression and therapy resistance including in CRC [7], and therefore are attractive therapeutic targets. Both intracellular and extracellular signals including metabolic stress regulate CSC self-renewal and plasticity [8]. Remarkably, CSCs lose self-renewal properties when removed from their microenvironment, which implies an essential role for microenvironment in directing their fate [9].

Autophagy, a catabolic process initiated in response to cellular and/or metabolic stress is a survival-promoting program [10,11]. Autophagy plays a context-specific role in tumorigenesis; however, in the majority of the oncogenic environments, autophagy promotes cancer progression and drug (therapy) resistance [12]. Recent studies have shown that inhibiting autophagy can restore the chemosensitivity in resistant cancer cells [13,14,15]. However, chloroquine (CQ) and its derivatives, currently used to interfere with autophagy by inhibiting lysosomal degradation of the autophagic cargo, have not proven sufficiently effective and specific in clinical trials. In this regard, recent studies have identified phosphatidylinositol 3-kinase (PIK3C3/VPS34 kinase), a key regulator of autophagy, as a specific and potent target for inhibiting autophagy, including in cancer cells [15,16,17]. However, the premise of inhibiting CSCs through interference with the PIK3C3/VPS34 function to prevent CRC malignancy remains ill understood.

In the current study, we provided data using a dynamic autophagy flux reporter and other biochemical assays that chemotherapy induces autophagy in the CSC enriched population of colon cancer. Notably, 5-FU (5-florouracil) is a key component of the colon cancer therapies; however, it is not effective against chemoresistant colon cancer. We further demonstrated using 36-077, a specific and potent inhibitor of PIK3C3/VPS34, that combinatorial therapy of 5-FU+36-077 augments efficacy of the 5-FU treatment by inhibiting CSC population in manners dependent on the Wnt/β-catenin signaling.

## 2. Materials and Methods

### 2.1. Cell Culture and Reagents

Caco2 and HCT116 cells were either purchased from ATCC (American Type Culture Collection, Rockville, MD, USA) or were available in our laboratory. At the UNMC core facility, cell lines are frequently tested for authenticity by genomic analysis. Cells were maintained in Dulbecco’s modified eagle’s medium (DMEM) or a RPMI-1640 medium, as appropriate, which were supplemented with 10% fetal bovine serum (FBS) and penicillin streptomycin (50 µg/mL). All reagents and antibodies used in this study are listed in the Appendix A.

### 2.2. Retroviral Infection in Caco2 and HCT116 Cells

HEK-293 cells producing retroviral particles encoding LC3-RFP-LC3ΔG were used for the analysis of autophagy flux. HEK-293 cells were grown in DMEM medium supplemented with 10% FBS and penicillin/streptomycin (50 µg/mL). Culture medium was collected after forty-eight hours and retroviral particles were concentrated using a Lenti-X Concentrator (Takara# 631231). For infection of the retroviral particles, concentrated viral particles were mixed in 10 mL of DMEM or RMPI medium supplemented with 10% FBS and polybrene (8 µg/mL). Cells were incubated with medium containing the retroviral particles at 37 °C. Forty-eight hours post-infection, cells were treated with the desired drugs and analyzed for the autophagy flux.

### 2.3. Autophagy Flux Measurement

In order to measure the autophagy flux, we used the GFP-LC3-RFP-LC3ΔG retroviral-based plasmid system. In this reporter system, LC3-RFP-LC3ΔG was cleaved by the endogenous ATG4 protein to yield an equimolar amount of GFP-LC3 and RFP-LC3. Autophagy induction led to the conjugation of GFP-LC3 to phosphatidylethanolamine (PE) on isolation membrane of autophagosome and degradation when delivered to the lysosome. Meanwhile, RFP-LC3ΔG remained in cytosol because of the mutation in the cleavage site of RFP-LC3 recombinant protein, serving as an internal control. Thus, measuring the ratio of GFP:RFP represented the dynamic autophagy flux.

### 2.4. Synthesis of the VPS34/PIK3C3 Inhibitor (36-077): 1-Cyclopropyl-3-(2-(methylthio) pyrimidin-4-yl) propan-2-one (*3*)

4-methyl-2-(methylthio) pyrimidine, **1**, (0.68 mL, 0.8 equiv.) was added dropwise at −10 °C to a solution of lithium diisopropyl amide (LDA, 2.0 M in THF) (4.15 mL, 1.5 equiv.) in tetrahydrofuran (THF, 10 mL). The reaction was allowed to reach rt over 45 min, followed by addition of 2-cyclopropyl-*N*-methoxy-*N*-methylacetamide, **2**, (1.0 g, 1.0 equiv.) in THF (5 mL) at −10 °C. After 15 min of the start of addition, the product was confirmed via TLC (5% EtOAC: Hexane) and the reaction was quenched with an ice-cold 1N HCl solution (30 mL). The product was extracted using EtOAC (25 mL × 3). The organic layer was dried over sodium sulfate, filtered, concentrated, and loaded on silica gel. Flash chromatography was performed (0–10% EtOAC in hexane). The product contaminated with starting material (roughly 30%) was obtained as a yellow liquid, and used without further purification (Y = 0.94 g, 75% yield. LCMS: R_T_ = 2.45 min. @ 215 and 254 nm, *m/z* = 223.1, 225.1 [M + H]^+^).

### 2.5. (Z)-1-cyclopropyl-4-(dimethylamino)-3-(2-(methylthio) pyrimidin-4-yl) but-3-en-2-one (*4*)

A solution of 1-cyclopropyl-3-(2-(methylsulfanyl) pyrimidin-4-yl) propan-2-one, **3**, (0.94 g) in DMF-DMA (5 mL) was stirred at 80 °C for 2 h and then the reaction was partitioned between water and EtOAC. The organic layer was separated, and the aqueous layer was back-extracted. The organic extracts were combined, washed with brine, dried over sodium sulfate, filtered, and concentrated. The crude product was purified by flash chromatography (25–100% EtOAC: Hexane). The product was obtained as a yellowish solid (yield = 0.94 g, 80%. LCMS: R_T_ = 2.20 min., >98% pure @ 215 and 254 nm, *m/z* = 278.0 [M + H]^+^).

### 2.6. 4′-(Cyclopropylmethyl)-2-(methylthio)-[4,5′-bipyrimidin]-2′-amine (*5*)

A solution of (*Z*)-1-cyclopropyl-4-(dimethylamino)-3-(2-(methylthio) pyrimidin-4-yl)but-3-en-2-one, **4**, (0.45 g, 1.0 equiv.), *N*-methylguanidine (0.23 g, 1.5 equiv.), and K_2_CO_3_ (0.89 g, 4.0 equiv.) in DMF (10 mL) was stirred at 120 °C for 3 h, followed by stirring at rt for 16 h. The reaction was dumped in 40 mL of ice cold water, and the product precipitated out. The product was isolated via filtration and then washed with water followed by hexane. The solid product was dried and used without further purification. The product was obtained as a white solid (yield = 0.32 g, 72%. LCMS: R_T_ = 2.37 min., >98% pure @ 215 and 254 nm, *m/z* = 274.0, 276.0 [M + H]^+^. ^1^H NMR (499 MHz, CDCl_3_) δ 8.53 (d, *J* = 5.1 Hz, 1H), 8.40 (s, 1H), 7.05 (d, *J* = 5.1 Hz, 1H), 5.35 (s, 2H), 2.82 (d, *J* = 6.8 Hz, 2H), 2.58 (s, 3H), 1.11–1.01 (m, 1H), 0.43 (d, *J* = 8.0 Hz, 2H), 0.15 (d, *J* = 5.1 Hz, 2H)).

### 2.7. 4′-(Cyclopropylmethyl)-2-(methylsulfinyl)-[4,5′-bipyrimidin]-2′-amine (*6*)

*m*-chloroperoxybenzoic acid (*m*-CPBA) (0.24 g, 1.2 equiv.) was added to a solution of 4′-(cyclopropylmethyl)-2-(methylthio)-[4,5′-bipyrimidin]-2′-amine, **5**, (0.32 g, 1.0 equiv.) in CH_2_Cl_2_ (10 mL) and the reaction mixture was stirred at 10 °C for 30 min, followed by the addition of another 0.5 equiv. of *m*-CPBA, which was stirred for an additional 15 min. The reaction was partitioned between water and CH_2_Cl_2_. The organic layer was separated, and the aqueous layer was back-extracted. The organic extracts were combined, washed with brine, dried over sodium sulfate, filtered, and concentrated. The crude product was purified by flash chromatography (0–10% MeOH in CH_2_Cl_2_). The product was obtained as a white solid (yield = 0.23 g, 67%. LCMS: R_T_ = 1.76 min., >98% pure @ 215 and 254 nm, *m/z* = 290.0, 292.0 [M + H]^+^. ^1^H NMR (499 MHz, DMSO) δ 8.95 (d, *J* = 5.3 Hz, 1H), 8.55 (s, 1H), 7.87 (d, *J* = 5.3 Hz, 1H), 7.19 (s, 2H), 2.90 (s, 3H), 2.83 (d, *J* = 6.8 Hz, 2H), 1.09 (ddd, *J* = 12.5, 7.5, 5.0 Hz, 1H), 0.36 (dd, *J* = 8.0, 1.5 Hz, 2H), 0.13 (d, *J* = 4.2 Hz, 2H)).

### 2.8. 4′-(Cyclopropylmethyl)-N^2^-(pyridin-4-yl)-[4,5′-bipyrimidine]-2,2′-diamine (*7*)

LiHMDS (1.56 mL, 3.0 equiv.) was added to a solution of 4-aminopyridine (146 mg, 3.0 equiv.) in THF (6 mL) at −10 °C and the reaction was warmed to rt and stirred for 30 min. The reaction was recooled to −10 °C and 4′-(cyclopropylmethyl)-2-(methylsulfinyl)-[4,5′-bipyrimidin]-2′-amine, **6**, (150 mg, 1.0 equiv.) was added. The reaction was slowly warmed up to rt, then the reaction was allowed to stir for 1 h. The reaction was quenched with saturated ammonium chloride solution, then concentrated in vacuo. The reaction was purified by a reverse-phase Gilson HPLC (0–100% ACN in water over 20 min). The product was obtained as a brown solid (yield = 10.0 mg, 6%. LCMS: 99% pure, R_T_ = 2.19 min.@ 215 and 254 nm, *m/z* = 320.1 [M + H]^+^. ^1^H NMR (499 MHz, DMSO) δ 8.60 (d, *J* = 5.1 Hz, 1H), 8.40 (s, 1H), 8.36 (d, *J* = 6.0 Hz, 2H), 8.16 (s, 1H), 7.78 (d, *J* = 6.3 Hz, 2H), 7.16 (d, *J* = 5.1 Hz, 1H), 6.99 (s, 2H), 2.81 (d, *J* = 6.9 Hz, 2H), 1.02–0.93 (m, 1H), 0.38–0.29 (m, 2H), 0.02 (q, *J* = 4.9 Hz, 2H)).

### 2.9. APCMin/+Colon Tumoroid Culture

Colon tumors from *APCMin/+* mice were chopped and resuspended in 1 mM of EDTA prepared in PBS. The resulting suspension were vigorously vortexed for 5–7 min to lose normal epithelial cells attached to the tumors and the remaining tumor chunks were digested by (0.1%) collagenase enzymes, as described [18]. Thereafter, suspension was passed through a 70-µM filter. The flowthrough was further centrifuged at 1500 RPM (10 min) and the pellet was washed with PBS × 2 time at 1500 RPM. All processes were done in sterile conditions and at 4 °C. Resulting tumor cells were embedded in a Matrigel bed in a 24-well culture dish and bathed in special cell culture medium containing N2 (50 µM) and B27(100 µM) as supplements, and EGF (100 ng/mL), as done for the 3D organ culture [19].

### 2.10. Western Blot Analysis

Western blot was done as described previously [20]. Briefly, cell or tissue lysates were subjected to sonication before centrifugation at 13,000× *g* (4 °C for 10 min). Protein quantification was done using the Bradford method (Bio-Rad # 5000001) and samples were prepared in 6× loading dye. Lysates were resolved using SDS-PAGE and the signal was visualized with horseradish peroxidase-conjugated secondary antibodies using enhanced chemiluminescence (Amersham Biosciences, Piscataway, NJ, USA # A38555).

### 2.11. Immunofluorescence

Immunostaining was done as previously described [20]. In brief, cells were fixed in 4% paraformaldehyde and then washed with phosphate buffered saline (PBS) containing 50 mM NH_4_Cl. Cells were then permeabilized with PBS containing 0.2% Triton X-100 (5 min) and incubated with blocking buffer (2% BSA + 2.5% normal goat serum) for 1 h at room temperature followed by incubation with an antigen-specific antibody (in blocking buffer) at 4 °C for 16–18 h. Subsequently, cells were washed with PBS (×5 times) buffer containing 0.1% Triton X-100 and incubated with secondary anybody conjugated with FITC, Cyc-3 or Rhodamine, as required. Images were captured using Nikon-T20 microscope.

### 2.12. Promoter Activity

For analysis of the Wnt-reporter activity, cells were transfected with TOP-FLASH reporter plasmid with phRL-TK (Promega Inc., Madison, WI, USA) plasmid and control plasmid constructs. Twenty-four hours post-transfection, cells were treated with different drugs and effects on reporter activity were analyzed. Transfection efficiency was normalized to Renilla luciferase activity of the phRL-TK. Results were expressed as the mean relative luciferase activity ± s.e.m.

### 2.13. RNA Isolation and qRT-PCR Analysis

RNA isolation and qRT-PCR were done as described previously [21]. Real-time PCR (qPCR) and gene specific primers are detailed in Appendix A. qPCR reactions were done using 25 ng of cDNA/reaction and 2×iQTM SYBR Green Super mix (Bio-Rad, Hercules, CA, USA) and for 25 cycles, unless described otherwise.

### 2.14. Isolation of Side Population (SP) Cells

Side population cells were isolated from DLD1 cell line. Cells were trypsinized by 0.25% trypsin-EDTA followed by centrifugation (1000 rpm × 5 min). Cells were then suspended in PBS containing 5% FBS and stained with Hoechst-33342 (Sigma# B2261-25MG; 10 µg/mL) with or without Verapamil (100 µM; V4629). Cells were then incubated for 90 min at 37 °C. During incubation, cells were shaken every 10 min for a pulse of 10–15 s. Cells were then centrifuged at 1000 rpm × 5 min and suspended in PBS with 5% FBS at 1 × 10^7^ cells/mL. Cells were then stained with propidium iodide (PI) for 5 min to remove the dead cells. The remaining cells were sorted using the flow cytometer. The Hoechst 33,342 dye was excited at 355 nm and its dual-wavelength fluorescence was analyzed (blue, 450 nm; red, 675 nm). SP cells were further cultured under standard embryonic stem cell conditions.

### 2.15. Statistical Analysis

Student’s *t*-test, Fisher exact test, and analysis of variance were used to determine statistical significance as applicable, and differences were considered statistically significant at *p* < 0.05. Results were plotted using Prism 9.0 (GraphPad Software, San Diego, CA USA) All data presented are representative of at least three repeat experiments and are presented as mean ± s.e.m. unless described otherwise.

## 3. Results

### 3.1. Chemotherapy Induces Autophagy in Colon Cancer

The 5-FU therapy remains the state-of-the-art chemotherapy option for treating colon cancer; however, its cancer cell killing activity is limited in the drug-resistant colon cancer [22]. To elucidate if autophagy helps CRC cells in developing resistance to chemotherapy, we employed a fluorescent reporter system (LC3/GFP/RFP) that measures autophagy in a dynamic fashion (details in Section 2). In brief, green fluorescent protein (GFP) expression coincides with autophagy inhibition, and red fluorescent protein (RFP) expression represents autophagy induction [23]. Colon cancer cells harboring this reporter system were cultured in the presence of 5-FU for twenty-four hours. The relative GFP/RFP expressions between control (vehicle-treated) and 5-FU treated cells were determined using Inverted fluorescence microscopy and puncta were quantitated. The 5-FU treatment led to a robust increase in LC3-RFP puncta (versus control cells) in both Caco-2 and HCT116 cells (Figure 1A,B). To validate that 5-FU treatment induced autophagy, we further determined the status of the established autophagy markers, P62/SQSTM1 and conversion of LC3-I (microtubule-associated proteins light chain-I) to LC3-II (microtubule-associated proteins light chain-II). Immunoblotting using cell lysates from control and 5-FU-treated cells showed a dose-dependent decrease in P62/SQSTM1 expression (*** *p* < 0.001) while the LC3-II/LC3-I ratio was significantly upregulated (*** *p* < 0.001) in the 5-FU-treated cells (Figure 1C–F). Similar changes in P62/SQSTM1 and LC3-II/I ratio in 5-FU-treated colon tumoroids (from the *APCMin/+* mice) (** *p* < 0.01 at 10 µM of 5-FU), supported the data from the colon cancer cells (Figure 1G,H). Taken together, our data suggested that the 5-FU-treatment induces autophagy in colon cancer.

### 3.2. Chemotherapy-Induced Autophagy Is Specific to Cancer Stem Cells

Cancer stem cells (CSCs) are central to the resistance to cancer therapy and are thought to be resistant to the chemotherapy-associated apoptotic stress (reviewed by Nguyen et al. and Beck et al. [24,25]). Hence, we determined if 5-FU treatment-induced autophagy was associated with the CSCs. Colon cancer cells expressing the autophagy reporter system, as in above studies, were subjected to 5-FU treatment for 24-h and then subjected to FACS-sorting to obtain GFP- and RFP-enriched cell populations (schematic representation in Figure 2A). The GFP-enriched population was significantly higher compared to the RFP positive cell population (Figure 2B,C; *** *p* < 0.001). We then performed qPCR analysis using total RNA isolated from the GFP- and RFP-enriched cells for cancer stem cell markers, LGR5, CD133, CD166, and OLFM4. As shown in Figure 2D, we found significant upregulation in LGR5 (*** *p* < 0.001), CD133 (** *p* < 0.01), CD166 (* *p* < 0.05), and OLFM4 (* *p* < 0.05) expressions in the RFP-positive cells compared to the GFP-positive cells. Taken together, these data suggested that chemotherapy-induced autophagy was limited to a subset of colon cancer cells with CSC characteristics.

### 3.3. 36-077, a PIK3C3/VPS34 Inhibitor, Is a Highly Specific and Potent Autophagy Inhibitor

Currently, hydroxychloroquine (HCQ) is the only FDA-approved drug that is known to inhibit autophagy. However, clinical trials using HCQ for improving cancer therapy have not been promising [26]. Recent studies have shown the potential of inhibiting VPS34-activity as a highly potent means for inhibiting autophagy [27]. Hence, based on the recently published reports focusing on VPS34 inhibition [28,29,30] we synthesized a highly potent and specific VPS34 inhibitor that was potent, selective and potentially effective in vivo [31], and named it 36-077. The chemical structure of 36-077 is presented in Figure 3A and the synthesis route has been described in the Appendix A and Section 2. We first validated the efficacy of 36-077 in inhibiting autophagy in colon cancer cells. Using Caco-2 cells, we determined the dose-response effects of 36-077 treatment upon autophagy. As shown in Figure 3B, accumulation of vacuoles, a cellular phenomenon associated with the inhibition of autophagy, was sharply upregulated in 36-077 treated cells compared to the vehicle-treated cells. Immunoblot analysis using total cell lysates further showed a significant and dose-dependent upregulation of P62/SQSTM1 expression in 36-077-treated cells (versus control cells; *** *p* < 0.001). The LC3-II/LC3-I ratio was sharply downregulated in same samples (Figure 3C,D; *** *p* < 0.001 at 10 or 50 nM). We found a similar dose-dependent effect of 36-077-treatment upon autophagy in the 3D-culture of colon tumors from *APCMin/+* mice (Figure 3E–G; *** *p* < 0.001 at 10 nM and 50 nM). Overall, the above data confirmed the efficacy of 36-077 in inhibiting autophagy in colon cancer cells and colon tumors.

### 3.4. Co-Treatment with 36-077 Improves Therapeutic Efficacy of 5-FU Treatment

Based on above findings, we postulated that the combination treatment with 36-077 would improve the efficacy of 5-FU therapy. To test, we treated CRC cells with 5-FU or 36-077, or in combination. The dose response effect was determined upon cell survival. As demonstrated in Figure 4A, co-treatment of HCT116 cells with 5-FU+36-077 markedly lowered the EC_50_ (relative global growth inhibition) for both 5-FU and 36-077 from 10.44 and 29.75 nM, respectively, to 4.19 nM. To determine whether the effects of above treatments are dependent on the changes in cell survival and/or apoptosis, we determined expression of cyclin-D1, a cell proliferation marker, and cleaved caspase-3, a marker of cell death. These studies were done using Caco-2, HCT116 cells, and colon tumoroids. The 5-FU treatment induced a dose-dependent decrease in cyclin-D1 expression, which was in line with its known effect in inhibiting the cell cycle progression (Figure 4B–G). However, 36-077, even at 50 nM, was only modestly effective in inhibiting cyclin-D1 expression or inducing the expression of cleaved caspase-3. The combination treatment (5-FU+36-077) not only resulted in a significant increase in the expression of cleaved caspase-3, but also a contrasting downregulation of cyclin-D1 expression (Figure 4B–G). Both cyclin-D1 inhibition and caspase-3 induction can inhibit cancer cell survival. We therefore calculated the “cell proliferation index”, the ratio of cell proliferation versus apoptosis for an unbiased view of the potential impact of the changes in these proteins upon cell survival. The ratio of cyclin-D1/caspase-3 expressions showed that the co-treatment of 36-077 significantly improved the therapeutic efficacy of 5-FU (*** *p* < 0.001 at 5-FU (10 µM) and 36-077 (50 nM)).

### 3.5. Co-Treatment of 5-FU and 36-077 Inhibits GSK-3β/Wnt/β-Catenin Signaling

We further examined how 36-077 co-treatment potentiates the efficacy of 5-FU therapy. Based on an initial screening of oncogenic signaling pathways, we focused on Wnt/β-catenin signaling considering its central role in CRC and in promoting the CSC niche [32]. To analyze the status of Wnt/β-catenin signaling, we used the highly sensitive TOP-FLASH reporter assay, as described previously [20]. Both 5-FU and 36-077-treatments significantly inhibited the TOP-FLASH reporter activity; however, combination treatment inhibited the reporter activity to its nadir (Figure 5A; **** *p* < 0.0001). GSK-3β regulates the Wnt/β-catenin signaling and destabilizes β-catenin expression by phosphorylating the protein at positions Ser33/Ser37/Thr41 [33,34,35]. We therefore determined if 5-FU+36-077 treatment affected the β-catenin phosphorylation (Ser33/Ser37/Thr41) to regulate Wnt-signaling. Immunoblotting was done using the total cell lysates and phospho-specific antibody, which demonstrated that the co-treatment of 5-FU and 36-077 significantly upregulated *p*-β-catenin (Ser33/Ser37/Thr41) expression (*** *p* < 0.001) (Figure 5B–D). Accompanying changes in the β-catenin protein expression supported its degradation. Based on the above findings, we examined if GSK-3β expression and phosphorylation (Ser9) were modulated by the 5-FU+36-077 treatment (Figure 5B–D). Immunoblotting using the *p*-GSK-3β (Ser9) and GSK-3β antibodies showed that the combinatorial treatment also inhibited expression of *p*-GSK-3β (Ser9) (*** *p* < 0.001) (Figure 5B–D). These findings were consistent amongst the colon cancer cells and colon tumoroids (Figure 5B–D). Immunolocalization studies for β-catenin cellular localization further demonstrated aggregate formation and membrane-localized β-catenin expression in CRC cells treated with 5-FU+36-077 compared to the 5-FU or 36-077 treatment (Figure 5E). Based on the above findings, we proposed that the effects of the co-treatment with 36-077 in promoting the efficacy of 5-FU-treatment should be overcome by forced activation of the Wnt/β-catenin signaling. Of note, Wnt-3A is a ligand of the Wnt-signaling [36,37]. Caco-2 cells were treated with 5-FU or 36-077 or in combination and in the presence or absence of Wnt-3A (10 nM). Immunoblotting using the cell lysates from the respective cell lines demonstrated an expected downregulation in the expression of *p*-β-catenin (Ser33/Ser37/Thr41) in cells treated with 5-FU+36-077 + Wnt-3A compared to the cells that received 5-FU+36-077 (Figure 5F,G; *** *p* < 0.001). We also found robust increases in cyclin-D1 expression in these cells compared to the cells receiving only 5-FU+36-077 (Figure 5F,G, ** *p*< 0.01). The cleaved caspase-3 expression was also significantly downregulated in cells receiving the 5-FU+36-077 + Wnt-3A compared to the cells receiving 5-FU+36-077. Wnt signaling regulates the CSC phenotypes, which are resistant to the anti-cancer therapy. Therefore, we further analyzed the expression of CSCs markers in cells treated with 5-FU+36-077 + Wnt-3A. The qPCR analysis demonstrated significant (*** *p* < 0.001) upregulation of the CSC markers in cells receiving 5-FU+36-077 + Wnt-3A versus 5-FU+36-077 (Appendix A). Taken together, our data suggested that inhibiting autophagy using 36-077, a PIK3C3/VPS34 inhibitor, along with 5-FU treatment, has a profound effect on the GSK-3β/Wnt/β-catenin signaling to inhibit cancer stem cells.

### 3.6. Combination Treatment of 5-FU+36-077 Inhibits Survival of Cancer Stem Cells

Since the above data suggested that combination treatment of 5-FU+36-077 targets the CSC population, we further utilized the spheroid assay commonly used for CSC culture, as described in [38], to further test our hypothesis. Effects of 5-FU (10 µM) treatment alone or with 36-077 (50 nM) were examined. As shown in Figure 6A, 5-FU+36-077 treatment inhibited the growth of spheroids significantly more than 5-FU or 36-077 alone. A similar outcome arose upon treating the spheroid cultures generated using single cell preparation from the colon tumoroids (Figure 6B). Subsequent qPCR analysis for CSC markers (LGR5, CD166, and CD133) using total RNA isolated from the resultant spheroid cultures, further confirmed that combinatorial treatment was significantly more effective in inhibiting the CSC population than 5-FU or 36-077 treatments (Figure 6C).

To further validate, we used side population (SP) cells that we have isolated and established in culture from DLD-1 cells (Figure 6D and Section 2). Notably, SP cells are considered CSCs based on their characteristics including expression of CSC markers, and high level of tumorigenicity [39,40]. Immunoblot analysis of P62/SQSMT1 and LC3-I/LC3II expressions showed a significant upregulation of autophagy flux in SP cells compared to the non-side population DLD-1 cells (NSP) (Figure 6E). We treated the SP cells with 5-FU, 36-077, or the combination. Phase contrast images of SP cells treated for 48 h showed that the combination treatment significantly inhibited their colony formation ability compared to the control cells. (Figure 6F,G). Immunoblotting of the total cell lysate demonstrated a significant decrease in OLFM4, a stem cell marker, and survivin, a proliferation marker in cells receiving 5-FU+36-077 compared to their individual treatment (Figure 6H,I). The proliferative index (CyclinD1/Cleaved caspase-3 expression) was also sharply downregulated in the cells receiving 5-FU+36-077 treatment compared to their individual treatment (*** *p* < 0.001) (Figure 6H,I). In same samples, expression of *p*-β-catenin (Ser33/Ser37/Thr41) expression was significantly upregulated while *p*-GSK3-β (Ser9) expression was sharply downregulated. Taken together, these data demonstrated that co-treatment of 36-077 improves efficiency of 5-FU-treatment by inhibiting the CSC population by modulating the GSK-3β-catenin signaling.

### 3.7. 36-077 Treatment Increased the Therapeutic Efficacy of 5-FU by Inhibiting the WNT Signeling in CSC

In summary, heterogenous populations of cancer cells represent the differential autophagy flux. CSC represent high autophagy flux and treatment of 5-FU leads to induction of cells death in cancer cells except CSC. These cells have self-renewal properties and upregulated WNT signaling. These properties make the CSC resistance to 5-FU and are responsible for cancer reoccurrence. Treatment of 36-077 in combination with 5-FU significantly inhibits WNT signaling pathway and induces the cells death in CSC.

## 4. Discussion

Colon cancer is one of the leading causes of mortality worldwide and thus discovery of new drugs that can target the cancer stem cells, central to the limited efficacy of current therapeutics, is the focus of the current research. Our current findings address this knowledge gap and the central findings from our current work are: (1) Autophagy helps protect colon cancer stem cells against 5-FU treatment by modulating the GSK-3β/Wnt/β-catenin signaling; and (2) targeting VPS34/PIK3C3 can effectively kill the autophagy-high CSCs to promote efficacy of colon cancer therapies. Our findings are supported by the findings of other studies, suggesting a similar effect of inhibiting PIK3C3 in promoting anti-cancer therapy [13]. Remarkably, in our studies, 5-FU treatment, a key component of the CRC chemotherapy, induced autophagy only in a subset of cancer cells as determined using a dynamic autophagy flux reporter and FACS sorting. The robust expression of the CSC markers only in the RFP-enriched cells supported their CSC phenotype, which was further validated by the studies using spheroid cultures and the SP isolation. Overall, the current study provided conclusive evidence that inhibiting PIK3C3/VPS34 can help promote the efficacy of conventional colon cancer therapy. Other studies have tried similar approaches where pharmacological targeting of the key signaling pathways including Wnt-, Notch-, and Hippo/YAP-signaling has been used alone or in combination therapy to target the cancer stem cells [41,42,43].

A role for autophagy in oncogenesis including colon cancer progression is well documented, though the effects can be context dependent [13,44,45]. In this regard, autophagy can be cancer suppressive or promoting [46]. Nevertheless, recent studies have demonstrated that cancer cells use autophagy to overcome metabolic stress inflicted by the tumor growth as well as anti-cancer therapy [47]. These findings have led to the postulation that autophagy helps cancer cells resist cancer therapy by promoting the CSC population [48]. However, an oncogenic function for autophagy has generally been aligned with specific oncogenic status of the cancer cells and/or genetic mutations [49]. In this regard, studies have suggested that cancer cells harboring K-ras mutation are autophagy addicted [49,50]. While our findings support the role of the autophagy in promoting the resistance to chemotherapy, they also suggest that autophagy is an essential property of colon cancer cells irrespective of the cancer stage or genetic mutations. In this regard, we had consistent outcomes in CRC cells with differing genetic mutations and malignant potential, as well as the colon tumoroids from *APCMin/+* mice. Our findings are supported by studies that demonstrate autophagy is upregulated in *APCMin/+* mice tumors, and colon cancer cells [51].

Recent studies have reported multiple pharmacological means for inhibiting autophagy for potentiating the anti-cancer therapy [52]. The most widely used agent for this purpose is chloroquine, also the subject of several clinical trials [53]. However, these clinical trials have not resulted in anticipated success. The key hindrances can be the non-specificity of this chemical for inhibiting autophagy and/or associated side-effects with long-term use [54]. These limitations led to the recent focus on targeting the PIK3C3/VPS34, due to its key role in autophagy initiation [49]. Several inhibitors targeting VPS34 have been designed and tested for their efficacy in inhibiting autophagy [28,55]. The autophagy inhibitor 36-077 used in the current study was synthesized based on a recent report, where it was tested for its anti-cancer effects using renal cancer cells [28]. Our study not only validates the utility of this inhibitor in inhibiting autophagy in colon cancer but also provide data that it could be employed for translational use, in near future. The data showing that 36-077 treatment alone does not cause extensive cell death attests to the fact that this inhibitor may not be toxic, a major limitation in developing new cancer therapies. In this regard, another PIK3C3/VPS34 inhibitor was used in vivo over an extended period for an improved health outcome in mice compared to respective control mice. A recent report using yet another PIK3C3/VPS34 inhibitor showed similar anti-cancer effects in treating colon cancer by modulating the immune checkpoint regulators [16]. Our data suggest that 36-077 treatment may specifically target the cancer stem cells in co-treatment with 5-FU. Taken together, it appears that inhibiting PIK3C3/VPS34 may significantly boost the anti-colon cancer therapy though additional studies using murine models of CRC progression and therapy resistance are needed and are part of our ongoing investigations. However, considering the integral role of the autophagy in normal cellular homeostasis, it is important to determine the dose and duration for the use of the autophagy inhibitors in treating cancers. The use of normal intestinal organoids and 3D-crypt culture can be used for such determinations and are part of our ongoing studies.

Our data suggest that 36-077 promotes the efficacy of 5-FU treatment in killing colon cancer cells by modulating the GSK-3β/Wnt/β-catenin signaling, further emphasizing the translational potential of this inhibitor for improving colon cancer therapy. A role for the GSK-3β/Wnt/β-catenin signaling in promoting colon cancer malignancy including CSCs is well documented [56,57,58]. Recent studies have further demonstrated that different immune-oncogenic signaling ultimately activate the Wnt-signaling for promoting colon cancer [59]. While the precise molecular mechanism regarding how 36-077 affects GSK-3β/Wnt/β-catenin signaling when combined with 5-FU remains unclear, studies have suggested a dual role of β-catenin in autophagy and cell proliferation [60,61].

## 5. Conclusions

Taken together, in this report, we provided conclusive evidence that chemotherapy induces autophagy in a subset of colon cancer cells with high autophagy flux and CSC properties. We further provided data that 36-077, a highly potent and specific autophagy inhibitor, can boost the efficacy of conventional colon cancer therapy. We anticipate the outcome from this study to encourage translational studies adopting similar approaches for future clinical trials and clinical uses.

## Figures and Tables

**Figure 1 cancers-13-02168-f001:**
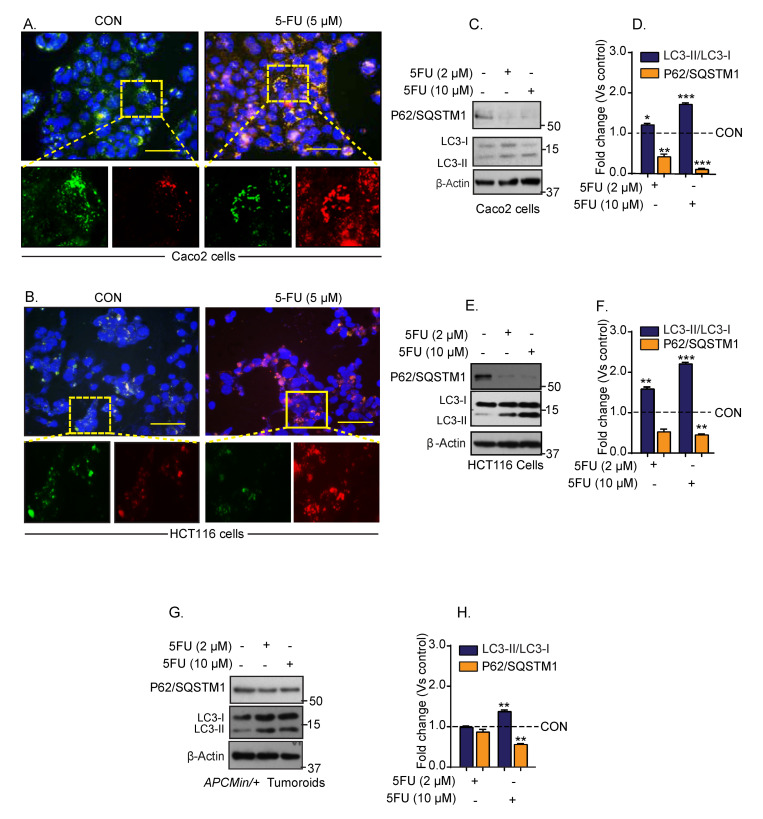
Conventional first-line colon cancer therapy drug 5-FU treatment induces autophagy in colon cancer cells. Caco-2 and HCT116 cells expressing GFP-LC3-RFP-LC3ΔG reporter expression construct were treated with 5-FU (5 µM) for 24 h. (**A**) Representative images of GFP- and RFP-positive Caco-2 cells; (**B**) representative images of the GFP- and RFP-positive HCT116 cells; (**C**–**H**) representative immunoblotting for P62/SQSTM1 and LC3-I and LC-II expressions, and densitometric analysis. Autophagy flux was evaluated by the densitometric analysis of LC3-II/LC3-I expressions. Statistical significance was determined by Student’s *t*-test. *** *p* < 0.001, ** *p* < 0.01, * *p* < 0.05. Scale bar = 50 μM.

**Figure 2 cancers-13-02168-f002:**
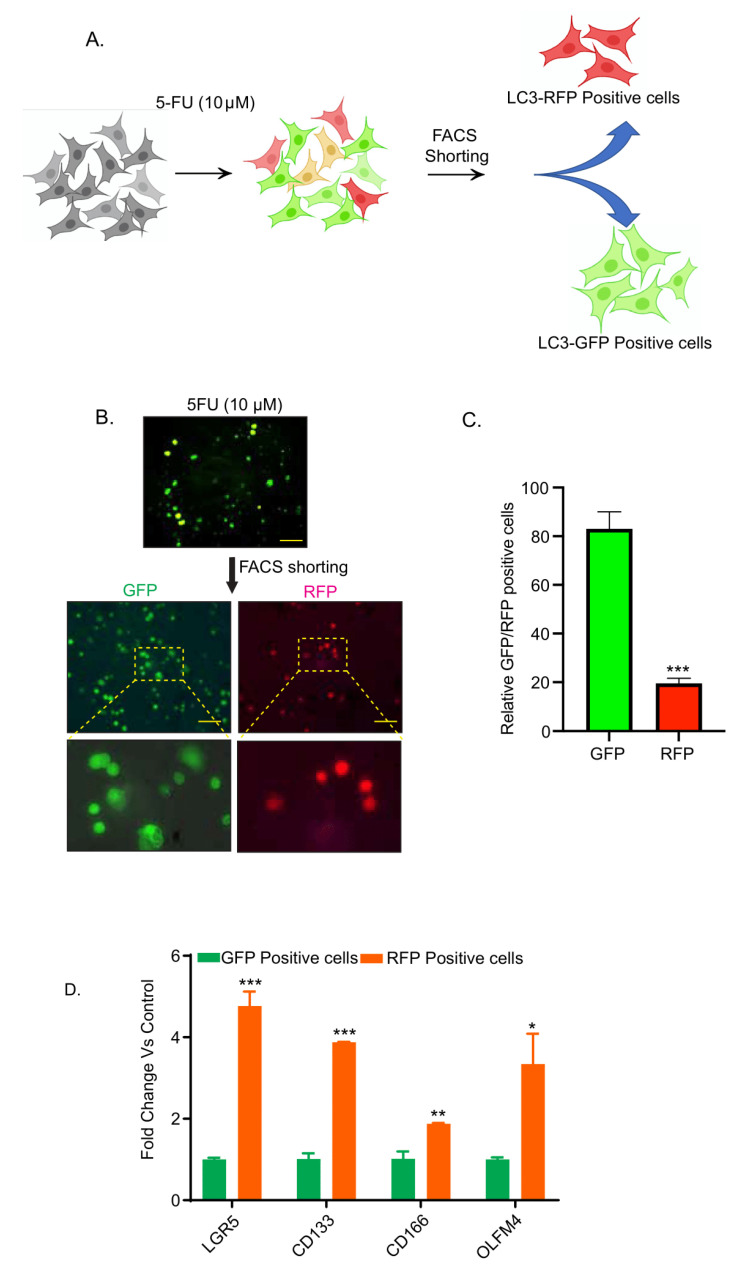
5-FU treatment-induced autophagy was limited to the colon cancer cells also enriched for the expression of cancer stem cell biomarkers. Caco-2 cells expressing GFP-LC3-RFP-LC3ΔG reporter expression construct were treated with 5-FU for 24 h and then subjected to FACS sorting. (**A**) The study design; (**B**) representative images of the FACS shorted autophagy-deficient (GFP) and autophagy-sufficient (RFP) cells; (**C**) relative ratio of GFP- and RFP-positive colon cancer cells in cells treated with 5FU; (**D**) qPCR analysis using total RNA from the RFP- and GFP-positive cells and specific CSC markers. *** *p* < 0.001, ** *p* < 0.01, and * *p* < 0.05. Scale bar = 100 µM.

**Figure 3 cancers-13-02168-f003:**
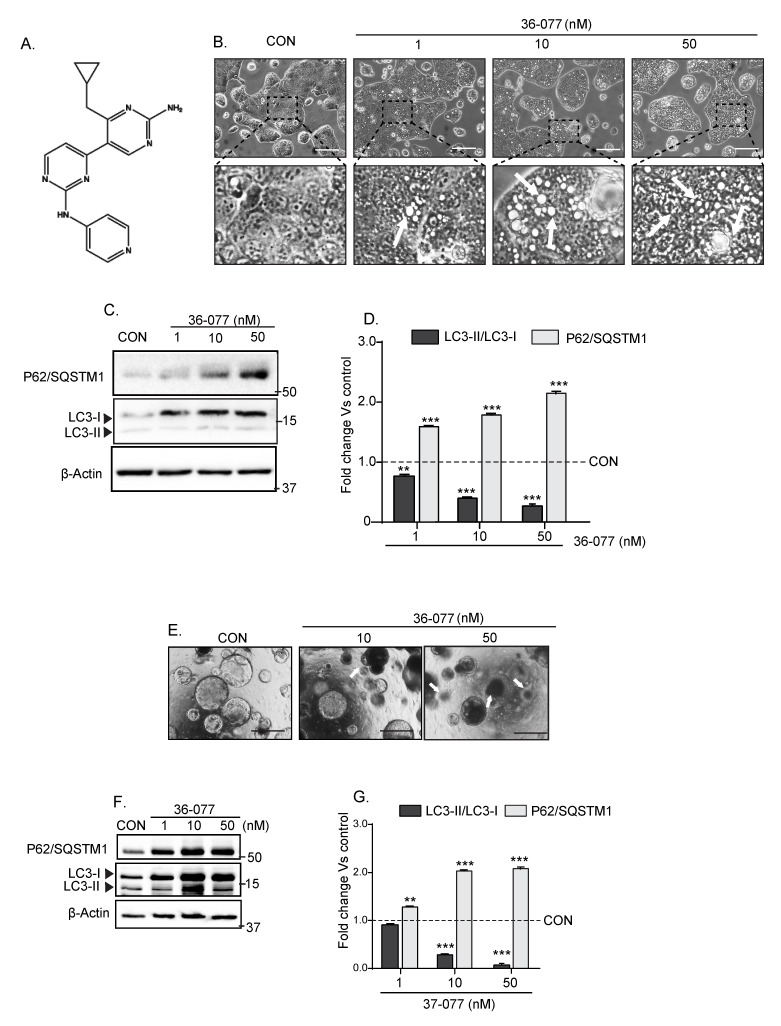
PIK3C3/VPS34 inhibitor 36-077 inhibits autophagy and induces cell death in colon cancer cells and 3D-cultured colon tumoroids. Caco-2 cells, HCT116 cells, or 3D-culture of the *APCMin/+* mice-derived colon tumors were treated with 36-077 and the status of autophagy flux was analyzed. (**A**) Chemical structure of 36-077; (**B**) phase-contrast images of Caco-2 cells subjected to 36-077 treatment (dose-response). Arrows indicate vacuoles; (**C**,**D**) representative immunoblot analysis for P62/SQSTM1 and LC3-I/LC3-II expression in Caco-2 cells subjected to 36-077 treatment (dose-response); (**E**) phase-contrast images of colon tumoroids subjected to 36-077 treatment (dose- response). Arrows indicate shrinking and dying tumoroids; (**F**,**G**) immunoblot analysis of P62/SQSTM1 and LC3-I/LC3-II expression using lysates from the control and 36-077-treated colon tumoroids. Statistical significance was determined by 1-way ANOVA. *** *p* < 0.001 and ** *p* < 0.01. Scale bar = 100 µM.

**Figure 4 cancers-13-02168-f004:**
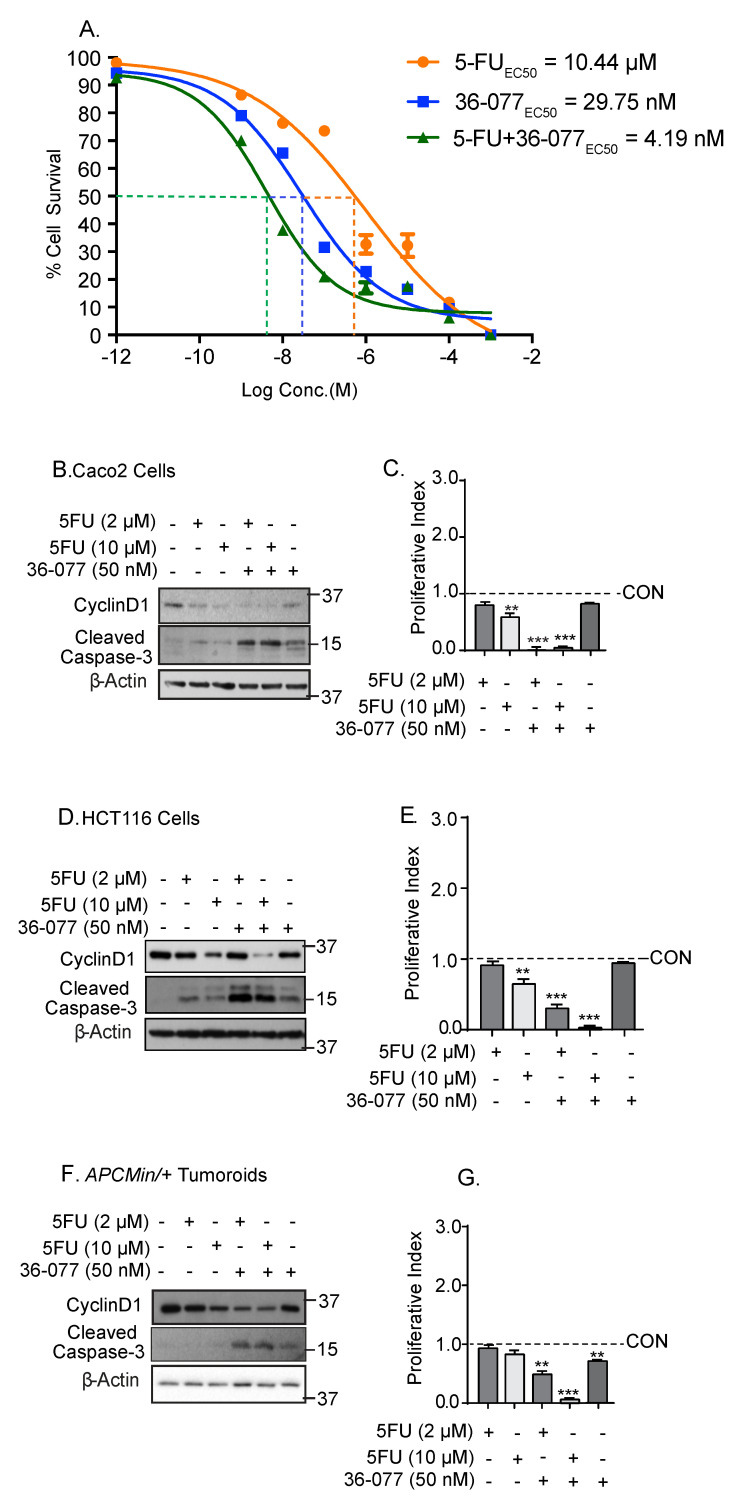
Combination treatment of 36-077 and 5-FU synergizes to inhibit colon cancer cell survival. Caco-2 and HCT116 cells were treated with 5-FU with or without 36-077 for 24 h. (**A**) Cell death was analyzed using cell viability assay. Effects of 5-FU and 36-077 alone, and in combination (HCT116 cells); (**B**–**G**) immunoblotting using total cell lysates from control and treated cells, and densitometry analysis. Caco-2 and HCT116 cells and colon tumoroids were used. Proliferative index was determined by calculating the ratio of cyclin-D1 and cleaved caspase-3 expression. Statistical significance was determined by 1-way ANOVA. Statistical significance was determined by Student’s *t*-test. *** *p* < 0.001 and ** *p* < 0.01.

**Figure 5 cancers-13-02168-f005:**
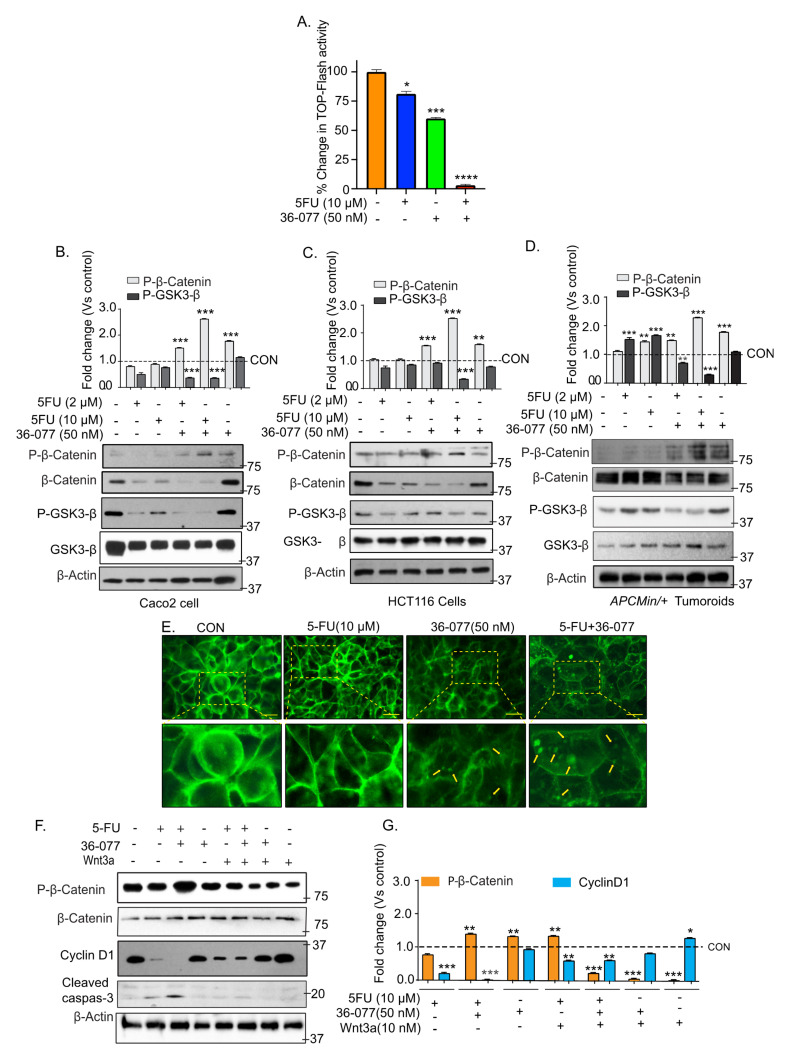
Combination therapy of 5-FU and 36-077 inhibits the WNT/β-catenin signaling. Caco-2 cells transfected with TOP-FLASH reporter expression vector along with Renilla plasmid and beta-catenin promoter activity was analyzed by luciferase assay. (**A**) TOP-FLASH activity in Caco-2 cells treated with 5-FU, 36-077 or both. Transfection efficiency of cells was normalized with Renilla expression; (**B**–**D**) representative immunoblots for *p*-β-catenin (Ser33/Ser37/Thr41) and *p*-GSK3β (Ser9) expression using the total cell lysates from Caco-2 cells, HCT116 cells, and colon tumoroids, respectively; (**E**) representative immunofluorescent analysis of β-catenin expression in Caco-2 cell treated with 5-FU with or without 36-077. Arrows indicate the aggregate formation and membrane localization; (**F**,**G**) immunoblotting and densitometry analysis of total cell lysate from Caco-2 cells treated with 5-FU+36-077 with and without WNT-3A, and densitometric analysis. Statistical significance was determined by 1-way ANOVA and Student’s *t*-test. *** *p* < 0.001, ** *p* < 0.01, * *p* < 0.05. Scale bar = 50 µM.

**Figure 6 cancers-13-02168-f006:**
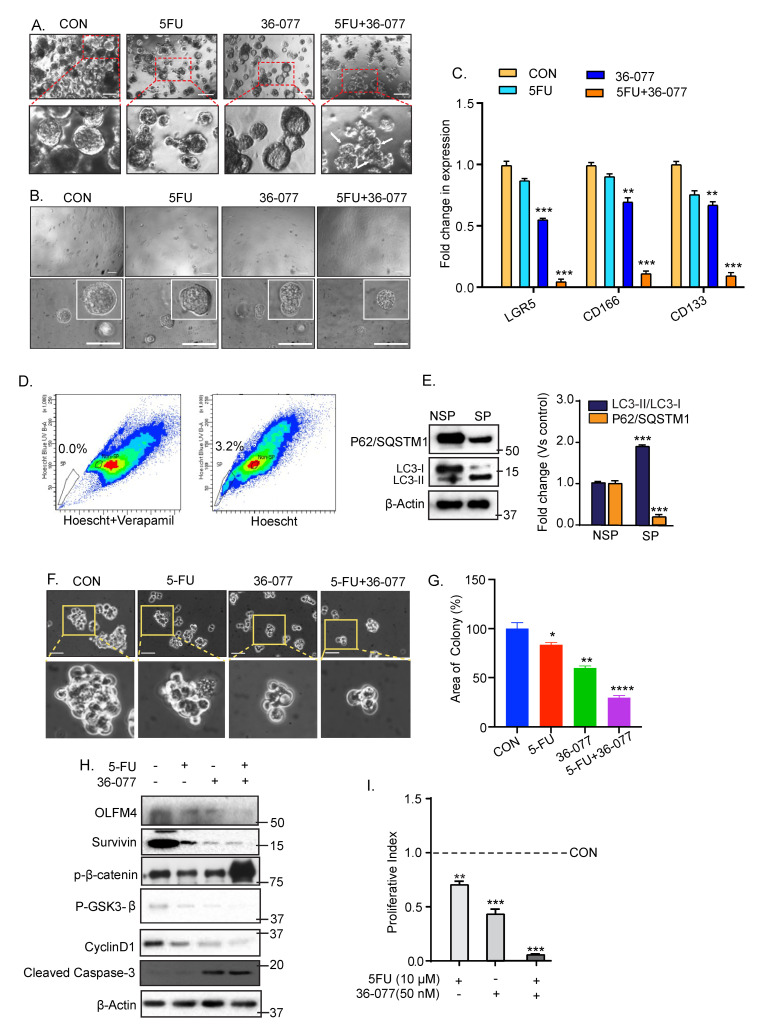
Combination treatment of 5-FU and 36-077 inhibits cancer stem cells. Single cells suspension of Caco-2 cells or *APCMin/+* mice-derived colon tumoroids were grown in stem cell culture medium for 6–7 days. Medium was changed every third or fourth day. Phase-contrast images of the resultant spheroids weres taken at day 7 of the culture. (**A**) Phase-contrast images of spheroid growth using Caco-2 cell treated with 5-FU with and without 36-077. Scale bar = 100 µM; (**B**) Phase-contrast images of spheroid formation using single cell populations prepared from colon tumoroids with 5-FU with and without 36-077. Scale bar = 100 µM; (**C**) qPCR analysis using RNA from spheroids of Caco2 cells treated 5-FU along with/without 36-077, for the determination of stem cells marker expression; (**D**) analysis of the side population (SP) cells in DLD1 cells. Cells were stained using Hoechst 33,342 dye along with or without verapamil and analyzed using flow cytometry. Cells treated with Hoechst and verapamil dye represent, SP cells (gated population) as a percentage (0.0%) of total cells in left panel (negative control) and cells treated with Hoechst dye only represent the SP cells as 3.2% of total population in right panel; (**E**) immunoblot analysis for P62/SQSTM1 and LC3-I/II expression using lysates from the side population (SP) or non-side population (NSP) cells; (**F**) phase-contrast images of SP cells cultured in stem cell culture medium and subjected to 5-FU treatment, with and without 36-077; (**G**) representative% change in the spheroid area of SP cells treated with 5-FU in combination with or without 36-077; (**H**) representative immunoblots for OLFM4, survivin, *p*-β-catenin (Ser33/Ser37/Thr41), *p*-GSK3β(Ser9), cyclinD1, and cleaved caspase-3; (**I**) densitometric analysis of proliferative index (CyclinD1/cleaved caspase-3). Statistical significance was determined by 1-way ANOVA and Student’s *t*-test. **** *p* < 0.0001 *** *p* < 0.001, ** *p* < 0.01, * *p* < 0.05.

## Data Availability

The data presented in this study are available in this article and Appendix A.

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
