# Peer review of "PIK3C3 Inhibition Promotes Sensitivity to Colon Cancer Therapy by Inhibiting Cancer Stem Cells"

_cancers, 2021, doi:10.3390/cancers13092168_

Round 1
Reviewer 1 Report
The manuscript is of high quality and deserves attention. Bioassays, pathways are properly treated. I have only one major comment that needs amendment. If this point is addressed, the manuscript can be accepted for publication.
- What i miss in the manuscript is the rationale to select 36-077. It is a PIK3C3/VPS34 inhibitor, but more details are needed to justify its selection and co-testing with FU. If possible, support this point with molecular modelling, or at least supply references that reinforce this point (reference to similar analogues or derivatives are acceptable).
- NMR data are needed for compounds 3 and 4.
Author Response
Q.1. What i miss in the manuscript is the rationale to select 36-077. It is a PIK3C3/VPS34 inhibitor, but more details are needed to justify its selection and co-testing with FU. If possible, support this point with molecular modelling, or at least supply references that reinforce this point (reference to similar analogues or derivatives are acceptable).
Response: We regret that reviewer felt that these rationales were not very clear. In the revised manuscript, we have further expanded these aspects for added clarity and also added additional references. In brief, we chose this inhibitor based on its potency, selectivity and in vivo efficacy in inhibiting autophagy. The co-testing of 36-077 with 5-FU was based on the fact that 5-FU is a key component of colon cancer therapies, as stated in the manuscript.
Q.2. NMR data are needed for compounds 3 and 4.
Response: The compound-3 was used immediately after its isolation. The NMR data for compound-4 is included in the “Materials and Methods” sections 2.4 and 2.5.
Reviewer 2 Report
The article is well written and presented a very robust sample and data. The methodology is very detailed and the discussion very clear.
Here are some specific comments:
(1) The graphical abstract has no legend; and also there is an error in the word "treatment". The authors had written "tretment";
(2) The authors should explain in more detail the importance of the use of 5-FU and why they had chosen this way.
(3) Also, the authors could explain in more detail why they will test the combination of 5-FU with 36-077;
(4) Point 2.10 - the authors began the text with minor letter. Capital letter should be used;
(5) Authors should pay attention to the formatting text;
(6) The statistical analysis is not well explained. The results of the tests performed could be shown in the article.
(7) Why is the reference to HCQ
is important for this study? The authors could explain better this fact.
(8) The authors write about the "CSC characteristics" but they don't explain which are them;
(9) The article had no conclusion.
Author Response
Q.1. The graphical abstract has no legend; and also there is an error in the word "treatment". The authors had written "tretment".
Response: We have updated the graphical abstract legend and corrected the spelling error.
Q.2 &3. The authors should explain in more detail the importance of the use of 5-FU and why they had chosen this way. Also, the authors could explain in more detail why they will test the combination of 5-FU with 36-077.
Response: The 5-FU was chosen because it is the key ingredient of colon cancer therapies. As mentioned in the manuscript, chemoresistance remains a key concern in the colon cancer management. Our data suggest that 5-FU treatment induces autophagy in cancer stem cells, which made the basis for the testing of the combination treatment of 5-FU and 36-077. We have attempted to further clarify these points in revised manuscript.
Q.4 &5. Point 2.10 - the authors began the text with minor letter. Capital letter should be used. Authors should pay attention to the formatting text.
Response: Thanks. We have made the suggested changes in the revised manuscript.
Q.6. The statistical analysis is not well explained. The results of the tests performed could be shown in the article.
Response: We regret that reviewer felt this way. We have included the details of statistical methods used for analysis and resulting data in the results and figure legends.
Q.7. Why is the reference to HCQ is important for this study? The authors could explain better this fact.
Response: Thanks. We have made minor changes to reflect that it is the Chloroquine and its derivatives. Moreover, the reference to HCQ is important because this is the only FDA drug known to inhibit autophagy however its use in clinical trial has not resulted in expected outcome. These limitations have led to the discovery of new autophagy inhibitors including the one used in current studies.
Q.8. The authors write about the "CSC characteristics" but they don't explain which are them.
Response: Since there are many excellent reviews focused on cancer stem cells, we remain focused primarily on the markers for the cancer stem cells.
Q.9. The article had no conclusion.
Response: We have added this section in the revised manuscript.
Reviewer 3 Report
The manuscript highlights the PIK3C3 inhibition inhibits promotes sensitivity to colon cancer therapy by inhibiting cancer stem cells, and the proposed strategy could offer a potential solution to clinical treatment of colon cancer therapy.
The quality of presentation of data and interpretation is excellent. It is recommended to accept the manuscript after a thorough check for spelling errors or typo such as....
In section 3.7, it is written WNT signeling several times instead of WNT signaling ..... 'CSC represent the represent high autophagy flux'...please delete the second 'represent' in this sentence...
with thanks and regards,
Author Response
Q.1 The quality of presentation of data and interpretation is excellent. It is recommended to accept the manuscript after a thorough check for spelling errors or typo such as....
Response: We appreciate reviewers positive comments and have done a through spell check in the revised manuscript.